# Effect of Different Tillage Practices on Sunflower (*Helianthus annuus*) Cultivation in a Crop Rotation System with Intercropping *Triticosecale*-*Pisum sativum*

**DOI:** 10.3390/plants11243500

**Published:** 2022-12-13

**Authors:** Aikaterini Molla, Georgios Charvalas, Maria Dereka, Elpiniki Skoufogianni

**Affiliations:** 1Laboratory of Soil Science, Department of Agriculture, Crop Production and Rural Environment, University of Thessaly, Fytokou Street, 38446 Volos, Greece; 2Laboratory of Development and Protection of Rural and Mountains Areas, Department of Planning and Regional Development, University of Thessaly, Pedion Areos, 38334 Volos, Greece; 3Department of Agrotechology, University of Thessaly, Gaiopolis, 41500 Larissa, Greece; 4Laboratory of Agronomy and Applied Crop Physiology, Department of Agriculture, Crop Production and Rural Environment, University of Thessaly, Fytokou Street, 38446 Volos, Greece

**Keywords:** LAI, SLA, N-uptake, total nitrogen, Greece

## Abstract

The objective of this work was to investigate the effect of different soil tillage practices on sunflower cultivation in a rotation system with intercropping of *Triticosecale-Pisum sativum*. For this purpose, a two-year experimental field with a 5% slope was established in central Greece. There were four treatments with three replications each. The treatments were as follows: (a) no tillage planting parallel to the contour (NTC-PAC), (b) conventional tillage planting parallel to the contour (CTC-PAC), (c) no tillage planting perpendicular to the contour (NTC-PEC), and (d) conventional tillage planting perpendicular to the contour (CTC-PEC). During the experiment, the plant height, leaf area index, specific leaf area, plants’ total nitrogen, and plants’ proteins were measured. According to the results, the plant height ranged from 64.9 (CTC-PAC) to 85.2 cm (NTC-PEC) for the first year and between 66.5–86.5 cm in for the CTC-PAC and NTC-PEC treatments in the second year. Furthermore, the leaf area index (LAI) and specific leaf area (SLA), plants’ total nitrogen and protein content and N-uptake were affected positively by the no tillage practice. To conclude, sunflower is a promising crop in a rotation system intercropping *Triticosecale*-*Pisum sativum*, cultivated under rainfed sloping conditions.

## 1. Introduction

Sunflower (*Helianthus annuus* L.) belongs to the *Compositae* family and is a native crop in North America. It was introduced into Europe in 1510 by Spanish explorers [1]. Additionally, sunflower is one of the most important new edible oil crops worldwide and it is also a desirable crop that can be satisfactorily cultivated in intensive dryland rotations [2]. Its oil has a high fatty acid content that is not synthesized by humans’ (linoleic and oleic acids) and it is used widely in the food industry in place of olive oil [3,4]. Furthermore, sunflower oil is used for biodiesel production, thus reducing the negative impacts of non-renewable energy sources in the environment and in society [5].

In many countries all over the world, the increase in fertilizer price in combination with the increasing cost of fossil fuels has caused producers to consider conservation agriculture, which includes lower impact cultivation practices such as no tillage cultivation, permanent crop residues on the surface, and crop rotation [6,7]. Conservation tillage is able to significantly improve soil properties (physical, biological, and chemical) and other biotic factors, reduce soil erosion, improve the water infiltration, and help in the reduction of the production costs [8,9,10].

Nitrogen (N) is the most essential nutrient in agricultural production. It plays an important role in crop photosynthesis and in the total biomass increase of the *Hellianthus annuus*. Zubillaga et al. [11] showed that the sunflower biomass increased from 19 to 40% when an optimum nitrogen amount was applied. In addition, nitrogen fertilizer had positive effects on the dry yield of sunflower according to other research [12,13]. On the other hand, nitrogen is considered to be one of the most expensive inputs in agricultural management [14].

It is well known that legumes have the ability to fix atmospheric N. Haugaard-Nielsen et al. [15] indicated that nitrogen uptake by legumes was higher in a rotation system compared with a monoculture system. Miller et al. [16] showed that pea fixes more atmospheric N than lentil and cowpea. *Pisum sativum* residues in a rotation system have beneficial effects on agronomic plant characteristics and on total crop production, wherefore the nitrogen requirements of the succeeding crops are reduced and less additional nitrogen fertilizer is applied [17]. According to Skoufogianni et al. [18], the incorporation of pea into a rotation system (pea–maize and pea–sunflower) increased the biomass and the nitrogen use efficiency.

The use of alternative cultural practices such as using no tillage (direct seeding) can be enhanced by the farmers. The no tillage practice is one of the three fundamental principles of conservation agriculture. This practice has better protection against soil erosion and offers a greater efficiency in nutrient uptake by plants [19,20].

The contour tillage is a more sustainable practice in comparison with that usually expected in flat fields (in straight lines) or for along-the-slope tillage. Adverse effects become more pronounced under intensive rainfall events. Contour cultivation on fields with a high inclination percentage can decrease soil erodibility, thus increasing topsoil resistance [21].

Information on the effect of cover crop using legumes for no tillage cultivation in combination with different planting systems (parallel and perpendicular to the contour) in sunflower cultivation in sloping land is lacking.

For this reason, we examined the effect of a two-year rotation system, including *Triticosecale-Pisum sativum* for winter cultivation and *Helianthus annuus* for summer cultivation, with different tillage practices (conventional and no tillage), being planted parallel or perpendicular to the contour on a dryland sloping field, on sunflower growth and quality under a Greek climate.

The aims of this work were to measure the plant height, leaf area index, specific leaf area, plants’ total nitrogen and protein content, and N uptake of sunflower cultivated under the above agricultural conditions.

## 2. Results

### 2.1. Meteorological Data

The meteorological data are presented in Figure 1. Total precipitation levels were 412.5 mm and 439 mm in 2015–2016 and 2016–2017, respectively. During the growing periods for sunflower (June–October), the precipitation was 226.6 mm in the first year and 187 mm in the second year. The higher rainfall events were in September and October. The temperature was at least 3 °C higher in the second year compared with the first year.

### 2.2. Plant Height

The plant height results are illustrated in Figure 2. The height ranged from 64.9 to 85.2 cm in the first year and from 66.5 to 86.5 cm in the second year. In all of the treatments, the height was higher in the first growing period compared with in the second year. Furthermore, the highest plant height was observed in the no tillage cultivation when planting parallel to the contour treatment for both of the studied growing seasons, followed by the conventional tillage parallel to the contour. Regarding the direction of the planting tillage (parallel and perpendicular to the contour), the tillage parallel to the contour increased the height of the plants, even though there were no statistically significant differences among the treatments. Additionally, the no tillage treatment resulted in the highest plant height each year, regardless of the tillage direction.

### 2.3. Leaf Area Index (LAI) and Specific Leaf Area (SLA)

The results of the leaf area index and specific leaf area are reported in Figure 3 and Figure 4. Three measurements of LAI and SLA were conducted, in the first year in 28 July 2015, 27 August 2015 and 15 September 2015, and in the second year in 18 July 2016, 16 August 2016 and 4 September 2016. The different tillage practices affected the LAI only in the second measurement, in both of the growing years. A statistically significant difference was observed between the NTC-PAD and CTC-PEC treatments. In the first and third measurements, there were no statistically significant differences between the treatments. In the two years of cultivation experiments, the maximum leaf area index (LAI) was recorded for the no tillage planting parallel to the contour direction (NTC-PAD) treatment. In the first year, the maximum values of LAI ranged from 3.6 (CTC-PEC) to 3.9 m^2^ m^−2^(NTC-PAC), and from 3.5 (CTC-PEC) to 4.01 m^2^ m^−2^ (NTC-PAC) in the second year. According to our results, the higher LAI values (3.9 the first year and 4.01 the second year) were observed when 134 (40 kg ha^−1^ from the fertilization plus 94 kg ha^−1^ from the incorporation of the *Triticosecale*-*Pisum sativum* cultivation) and 142 kg ha^−1^ (40 kg ha^−1^ from the fertilization plus 102 kg ha^−1^ from the incorporation of the *Triticosecale*-*Pisum sativum* cultivation) of nitrogen were applied in the first and second year, respectively. The tillage systems affected the specific leaf area, but only numerically. No statistically significant difference was observed between the different treatments. The mean values showed that the highest SLA was achieved in the no tillage planting parallel to the contour direction (NTC-PAC) treatment.

### 2.4. Total Nitrogen Content in Plants

Data regarding plants’ total nitrogen are presented in Table 1 and Table 2. In the first year, the highest total nitrogen (%) was observed in the NTC-PAC treatment (4.388%) and the lowest was in CTC-PEC (4.153%). In the second year, the mean values of total nitrogen ranged from 4.175 (CTC-PAC) to 4.435% NTC-PEC). In both cultivation years, a statistically significant difference was observed between NTC-PAC and the other treatments. The planting parallel to the contour showed better results compared with the planting perpendicular to the contour tillage in both years of experiments. In the first year (2015), the total nitrogen in the NTC-PAC treatment was higher by 3–5% in comparison with the other treatments. In the second year (2016), the increase in total nitrogen in the NTC-PEC ranged from 3 to 6%. Between the two cultivation years, the total nitrogen was better in the second year. The highest increase was noticed in the CTC-PAC treatment (2%). That total nitrogen increase was probably derived from the nitrogen residual of the previous cultivation year and from the *Triticosecale*-*Pisum sativum* residues that were incorporated in the second growing year. Furthermore, the rainfall in the time of incorporation during the second year accelerated the decomposition of the winter crop residues.

### 2.5. Protein Content

The results of the plants’ protein content (%) are illustrated in Table 3 and Table 4. The highest protein content was observed for the NTC-PAC treatment (27.43%) and the lowest was for CTC-PEC (25.97%). In the second year, the mean values of the proteins ranged from 27.72 (CTC-PAC) to 26.09% (NTC-PEC). In both cultivation years, a statistically significant difference was observed between the NTC-PAC and the other treatments. The planting parallel to the contour gave better results compared with the planting perpendicular to the contour tillage, for both of the experimental years. Between the two cultivation years, the protein contents were better in the second year. The highest increase was noticed for the CTC-PAC treatment (2%).

### 2.6. N-Uptake

Table 5 and Table 6 show the N-uptake for sunflowers as an undersown catch crop after pea cultivation in the two years studied, respectively. The highest value for N-uptake was observed in the no tillage planting parallel to the contour (NTC-PAC). In the first year, the NUE ranged from 156 to 235 kg ha^−1^ and a statistically significant difference was observed between the NTC-PAC, NTC-PEC, and CTC-PEC treatments. In the second cultivation year, the N-uptake values ranged from 192 to 265 kg ha^−1^. The results showed that between the NTC-PAC treatments and the other three ones, there was a statistically significant difference. In addition, an increase was noticed in the second year compared with the first year for all of the treatments. In the NTC-PAC treatment, the increase was at a rate of 11%. At this point, it should be mentioned that according to the results, the incorporation of pea residues could have positive effects on the increase in N-uptake over time.

## 3. Discussion

Our study showed that the plant height was higher for the no tillage practice. Our results are in disagreement with other investigations [22,23]. According to their study, the plant height was higher in conventional tillage in comparison with the no tillage cultivation. Sessiz et al. [22] found that the plant height ranged from 128 to 137 cm for conventional tillage and from 122 to 128 cm for the no tillage practice. In our research, the height of the plants varied from 64.9 to 80.1cm using conventional management and from 78.5 to 86.5 cm in the no tillage treatments. Furthermore, Mourad et al. [23] observed that the plant height was better in conventional tillage compared with no tillage, but in the second year of experiments, the height was lower in comparison with the first year in both cultivation practices. In our investigation, the results showed that in the second year, the plants’ height was higher. It is clear that after the second cultivation year, the incorporation of *Triticosecale*-*Pisum sativum* residues helped increase the final crop height.

Data from our study site indicate that the no tillage planting parallel to the contour had the highest LAI content. A statistically significant difference was only observed in the second measurement (27 August 2015 and 16 August 2016) between the different tillage practices in both of the growing years. Mujeed-ul-Haq et al. [13] noticed similar LAI values with our findings when 110 kg N ha^−1^ were applied in sunflower cultivation. In other research, the LAI reached a value of 5.13 and 5.37 m^2^ m^−2^ in conventional (reduced) and traditional tillage 90 days after sowing, respectively [23]. Aboudrare et al. [24] stated that the LAI was lower (2.39 m^2^ m^−2^) than our results using the no tillage cultivation. Only a few studies have addressed the relation between LAI and different tillage practices under sloping conditions for sunflower cultivation.

In our investigation, the plants’ total nitrogen ranged from 4.153% in the NTC-PEC treatment to 4.388% in the NTC-PAC during the first year. In the second year, the nitrogen values were higher than for the first year. Between the two growing years, the increase in nitrogen content ranged from 1 to 2%. Murillo et al. [25] found out that the total nitrogen content was from 0.76 to 3.48, which were lower values compared with our results.

In addition, the no tillage management had positive effects in the protein values compared with conventional practice. A statistically significant difference was observed in the second year of cultivation. Scheiner et al. [26] mentioned that the protein content was from 20.2 to 23.6%, which were lower values in comparison with our study.

Regarding N-uptake, the NTC-PAC treatment performed better with higher values compared with the other practices. The N-uptake reached values of 235 and 265 kg ha^−1^ in the first and second years, respectively. The same results have been mentioned in other studies [11,27]. The findings of Lopez-Bellino et al. [28] showed that the no tillage treatment positively impacted N-uptake, in agreement with our results. Skoufogianni et al. [18] found that the sunflower N-uptake reached a value of 221 kg ha^−1^ when pea was incorporated into the soil.

Our investigation indicated that pea residues incorporated into the field can provoke a significant increase in all of the above measurements.

## 4. Materials and Methods

### 4.1. Study Area

A field experiment with sunflower cultivation was conducted in the experimental station of the University of Thessaly (Larissa, Greece). The studied area, witha latitude of 39°37′30″ and a longitude of 22°22′51″, is located at an altitude of 80 m above sea level. Its climate is characterized as Mediterranean, with hot and dry summers as well as cold and wet winters.

### 4.2. Soil Analyses

A soil sample of the field was taken before the sowing period, each autumn, from a depth of 0–30 cm from the surface using a steel sampler. The soil sample was transported to the laboratory, and was air-dried and sieved through a 2 mm sieve. The soil was analyzed for pH (1:2.5 dH_2_O), electrical conductivity (1:5 dH_2_O), and calcium carbonate (CaCO_3_) using a calcimeter. The percentage (%) of sand, clay, and silt was measured using the Bouyoukos method. The organic matter was measured using the Walkley–Black method; the total nitrogen using the Kjeldahl method; the available soil P using the Olsen method, analyzed with ammonium vanadomolybdate/ascorbic blue and measured in a UV spectrophotometer at 882 nm; and the exchangeable Κ with 1:10 at 1 M CH_3_COONH_4_ pH 7, analyzed in a flame photometer. All of the analyses were carried out according to Rowell [29].

The soil was clay loam, with pH 8.21, organic matter between 1.6–1.65%, and total nitrogen of 0.08% in the first year and 0.085% in the second. The physicochemical properties of the soil are presented in Table 7.

### 4.3. Field Experiment

A two-year field experiment was established in a field with a slope of 5%. The experimental design was a split plot. The treatments were as follows: (a) no tillage cultivation planting parallel to the contour direction (NTC-PAD), (b) conventional tillage cultivation planting parallel to the contour direction (CTC-PAD), (c) no tillage cultivationplanting perpendicular to the contour direction (NTC-PED), and (d) conventional tillage cultivation planting perpendicular to the contour direction (CTC-PED).Each treatment had three replications. The plots were 132 m^2^ in size (6 m in width and 22 m in length). The planting arrangement was 75 × 20 cm^2^, consisting of eight rows 75 cm apart and an average plant density of sunflowers of 10 plants m^−2^.

Before the sowing, all of the necessary cultivation practices were conducted. The residues of the previous autumn cultivation at a rate of 30–70% (*Triticosecale-Pisum sativum*) were incorporated in the field at the end of their biological cycle and about 20 days before sunflower sowing. The mean total incorporated biomass of the intercropping was 2.4 and 2.63 t ha^−1^ for the first and second sunflower growing year, respectively. The procedure of the residue decomposition was accelerated by the natural rainfall. Moreover, conventional tillage plots were conducted by ploughing to a depth of about 25 cm. Sowing dates and other agronomic data are summarized in Table 8.

Fertilization was applied during sowing as the basal dressing using a compound fertilizer (N: 40 kg N ha^−1^, P: 60 kg P_2_O_5_ ha^−1^, and K: 60 kg K_2_O ha^−1^) in all of the plots. Furthermore, nitrogen was added in the soil from the previous cultivation (intercropping *Triticosecale*-*Pisum sativum*) due to the fixing of N_2_ from the atmosphere (Table 9). The plots were cultivated under rainfed conditions and this was because the specific area is considered a dryland one.

### 4.4. Experimental Measurements

#### 4.4.1. Plant Height

Ten plants from every plot were randomly selected. Plant height was measured using a measuring tape, from ground level to the top edge of the inflorescence.

#### 4.4.2. Leaf Area Index (LAI) and Specific Leaf Area (SLA)

To investigate the LAI and SLA of sunflower plants, three plants from each plot were randomly harvested early in the morning. Three measurements of LAI and SLA were conducted, for the first year on 28 July 2015, 27 August 2015 and 15 September 2015, and for the second year on 18 July 2016, 16 August 2016 and 4 September 2016. The leaf area index was determined using an automatic LI-COR, made in Nebraska (USA) and bought from HELLAMCO AE (Greece) (model LL-3000A). The specific leaf area and leaf area index were measured as the product of the green leaf area dry weight.

The connection between SLA and LAI is indicated by the equation:LAI = SL × SLA × 10^−4^(1)
where SL is the dry weight of the fresh leaves (kg ha^−1^).

#### 4.4.3. Total Nitrogen and Proteins in Plants

The plant samples from each plot after the harvest were transported in the Lab and remained in the oven at 70 °C until reaching a constant weight. The total nitrogen content was calculated from the whole plant biomass and the measurement was conducted using the Kjeldahl method [30].

The protein content was measured using the follow mathematical equation.
Protein content = total nitrogen content × 6.25(2)

#### 4.4.4. N-Uptake

The measurement of the N-uptake was performed using the equationbelow:N-uptake = yield × plant’ total nitrogen(3)

### 4.5. Statistical Analysis

The experimental data were analyzed using the statistical package Statgraphics plus 8.1 for the LSD test with a level of significance of about 95% (*p* < 0.05).

## 5. Conclusions

In this research, we evaluated the impact of no tillage on plant height, LAI, SLA, plants’ nitrogen, and proteins of sunflower in a rotation system intercropping *Triticosecale*-*Pisum sativum* under natural rainfall in comparison with conventional agriculture. In addition, we tested the effect of the planting direction (parallel and perpendicular to the contour).

The results showed that plant height was higher in the no tillage planting parallel to the contour cultivation. LAI (3.69 the first year and 4.01 the second year) had the maximum values in the NTC-PAC treatment. No tillage planting parallel to the contour direction (NTC-PAC) treatment, also had a positive impact on the SLA values.

The total nitrogen and proteins of the sunflower plants were better in the second year of the experiments and the highest increase was found for the CTC-PAC treatment (2%). Furthermore, the no tillage practice affected the N-uptake in both of the cultivation years.

To sum up, in a sloping dryland field, sunflower can be a promising crop when cultivated in a rotation system intercropping *Triticosecae* – *Pisum sativum* using the no tillage planting parallel to the contour processing in a Greek climate.

The incorporation of leguminous residues into fields can significantly increase the nitrogen content and many abandoned sloping lands could be cultivated.

## Figures and Tables

**Figure 1 plants-11-03500-f001:**
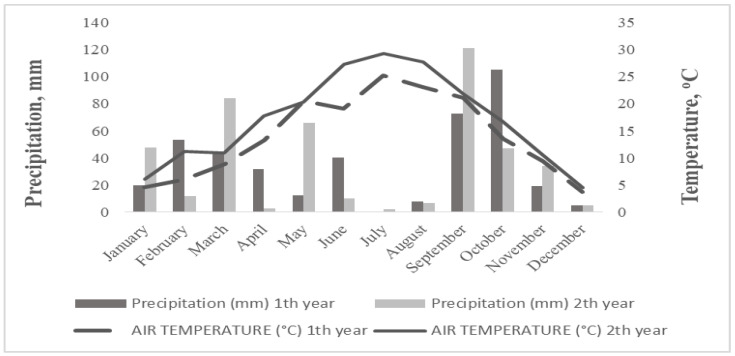
Average temperature (°C) and precipitation (mm) in the studied area during the 2015–2016 and 2016–2017 cultivation years.

**Figure 2 plants-11-03500-f002:**
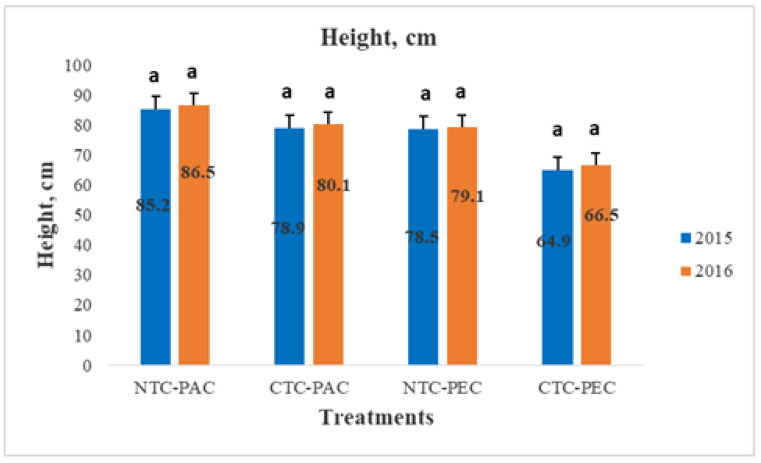
Plant height (cm) of sunflower cultivation for the two growing years. Different letters at each column denote statistically significant difference of means according to the LSD test for 95% significance level (*p* < 0.05). NTC-PAC: no tillage planting parallel to the contour direction. CTC-PAC: conventional tillage planting parallel to the contour direction. NTC-PEC: no tillage planting perpendicular to the contour direction. CTC-PEC: conventional tillage planting perpendicular the contour direction.

**Figure 3 plants-11-03500-f003:**
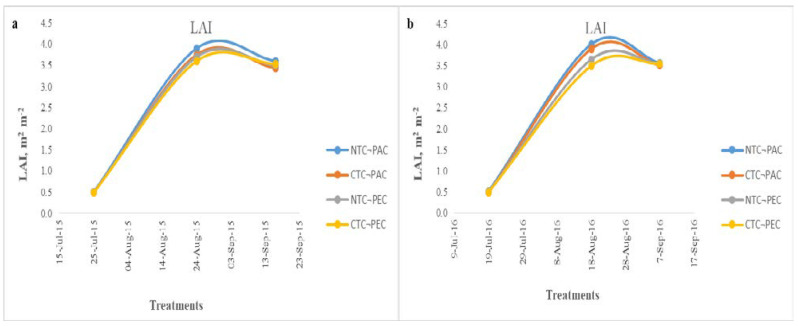
Leaf area index (LAI) of the sunflower cultivation in two growing years of (**a**) 2015–2016 and (**b**) 2016–2017. NTC-PAC: no tillage planting parallel to the contour direction. CTC-PAC: conventional tillage planting parallel to the contour direction. NTC-PEC: no tillage planting perpendicular to the contour direction. CTC-PEC: conventional tillage planting perpendicular the contour direction.

**Figure 4 plants-11-03500-f004:**
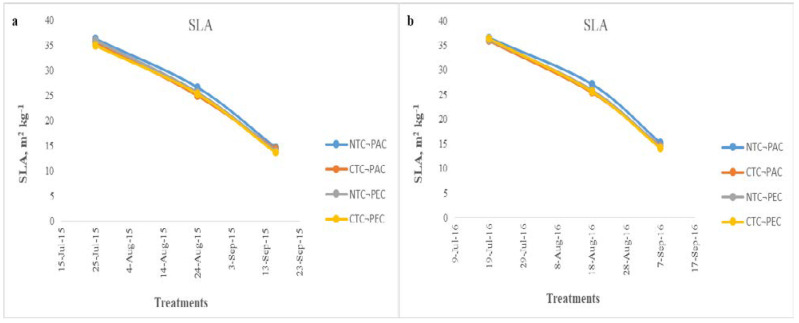
Specific leaf area (SLA) of the sunflower cultivation in two growing years of (**a**) 2015–2016 and (**b**) 2016–2017. NTC-PAC: no tillage planting parallel to the contour direction. CTC-PAC: conventional tillage planting parallel to the contour direction. NTC-PEC: no tillage planting perpendicular to the contour direction. CTC-PEC: conventional tillage planting perpendicular the contour direction.

**Table 1 plants-11-03500-t001:** Total nitrogen (%) of the *Helianthus annuus* cultivation for the 4 different soil practice treatments in the first growing year.

	Total Nitrogen, %		CV%
Treatment	*Helianthus annuus*		
NTC-PAC	4.388	c	8.15
CTC-PAC	4.251	b	3.22
NTC-PEC	4.250	b	1.76
CTC-PEC	4.153	a	6.15
LSD		0.0263	

Different letters at each column denote statistically significant difference of means according to the LSD test for 95% significance level (*p* < 0.05). NTC-PAC: no tillage planting parallel to the contour direction. CTC-PAC: conventional tillage planting parallel to the contour direction. NTC-PEC: no tillage planting perpendicular to the contour direction. CTC-PEC: conventional tillage planting perpendicular the contour direction.

**Table 2 plants-11-03500-t002:** Total nitrogen (%) of the *Helianthus annuus* cultivation in the 4 different soil practice treatments in the second growing year.

	Total Nitrogen, %		CV%
Treatment	*Helianthus annuus*		
NTC-PAC	4.435	c	6.75
CTC-PAC	4.320	b	2.25
NTC-PEC	4.310	b	8.55
CTC-PEC	4.175	a	4.31
LSD		0.0245	

Different letters at each column denote statistically significant difference of means according to the LSD test for 95% significance level (*p* < 0.05).NTC-PAC: no tillage planting parallel to the contour direction. CTC-PAC: conventional tillage planting parallel to the contour direction. NTC-PEC: no tillage planting perpendicular to the contour direction. CTC-PEC: conventional tillage planting perpendicular the contour direction.

**Table 3 plants-11-03500-t003:** Protein content (%) of the *Helianthus annuus* cultivation in the 4 different soil practice treatments in the first growing year.

	Protein Content, %		CV%
Treatment	*Helianthus annuus*		
NTC-PAC	27.43	c	8.15
CTC-PAC	26.57	b	3.22
NTC-PEC	26.56	b	1.76
CTC-PEC	25.97	a	6.15
LSD		0.1642	

Different letters at each column denote statistically significant difference of means according to the LSD test for 95% significance level (*p* < 0.05).NTC-PAC: no tillage planting parallel to the contour direction. CTC-PAC: conventional tillage planting parallel to the contour direction. NTC-PEC: no tillage planting perpendicular to the contour direction. CTC-PEC: conventional tillage planting perpendicular the contour direction.

**Table 4 plants-11-03500-t004:** Protein content (%) of the *Helianthus annuus* cultivation in the 4 different soil practice treatments in the second growing year.

	Protein Content, %		CV%
Treatment	*Helianthus annuus*		
NTC-PAC	27.72	c	6.75
CTC-PAC	27.00	b	2.25
NTC-PEC	26.94	b	8.55
CTC-PEC	26.09	a	4.31
LSD		0.1523	

Different letters at each column denote statistically significant difference of means according to the LSD test for 95% significance level (*p* < 0.05).NTC-PAC: no tillage planting parallel to the contour direction. CTC-PAC: conventional tillage planting parallel to the contour direction. NTC-PEC: no tillage planting perpendicular to the contour direction. CTC-PEC: conventional tillage planting perpendicular to the contour direction.

**Table 5 plants-11-03500-t005:** N-uptake of the *Helianthus annuus* cultivation in the 4 different soil practice treatments in the first growing year.

	N-Uptake, kg ha^−1^		CV%
Treatment	*Helianthus annuus*		
NTC-PAC	235	c	2.76
CTC-PAC	222	bc	8.17
NTC-PEC	210	b	6.55
CTC-PEC	156	a	3.69
LSD		6.4528	

Different letters at each column denote statistically significant difference of means according to the LSD test for 95% significance level (*p* < 0.05).NTC-PAC: no tillage planting parallel to the contour direction. CTC-PAC: conventional tillage planting parallel to the contour direction. NTC-PEC: no tillage planting perpendicular to the contour direction. CTC-PEC: conventional tillage planting perpendicular the contour direction.

**Table 6 plants-11-03500-t006:** N-uptake of the *Helianthus annuus* cultivation in the 4 different soil practice treatments in the second growing year.

	N-Utake, kg kg^−1^		CV%
Treatment	*Helianthus annuus*		
NTC-PAC	265	c	9.15
CTC-PAC	231	b	2.45
NTC-PEC	217	b	7.65
CTC-PEC	192	a	4.55
LSD		4.5947	

Different letters at each column denote statistically significant difference of means according to the LSD test for 95% significance level (*p* < 0.05).NTC-PAC: no tillage planting parallel to the contour direction. CTC-PAC: conventional tillage planting parallel to the contour direction. NTC-PEC: no tillage planting perpendicular to the contour direction. CTC-PEC: conventional tillage planting perpendicular the contour direction.

**Table 7 plants-11-03500-t007:** Physicochemical properties of the soil used.

Cultivation Year	pH	E.C. (μS cm^−1^)	CaCO_3_	Organic Matter (%)	Total Nitrogen (%)	Olsen P (mg kg^−1^)	Exchangeable Κ (mg kg^−1^)	Sand (%)	Clay (%)	Silt (%)
2015	8.21	435	16.5	1.65	0.08	21.24	216.06	38.41	36.11	25.48
2016	8.1	454	15.6	1.6	0.085	6.8	198.5	39.63	36.5	23.88

**Table 8 plants-11-03500-t008:** Agronomic data of sunflower cultivation.

	LARISSA	
	2015	2016
Incorporation date	8 June 2015	25 May 2016
Date of sowing	30 June 2015	12 June 2016
Date of flowering	20 August 2015	10 August 2016
LAI—SLA measurement	28 July 2015, 27 August 2015, 15 September 2015	18 July 2016, 16 August 2016, 4 September 2016
Date of harvest	17 October 2015	16 October 2016

**Table 9 plants-11-03500-t009:** N uptake when intercropping *Triticosecale*-*Pisumsativum* cultivation (kg N ha^−1^).

Treatments	2015			2016		
	Yield	Total Nitrogen	N-Uptake	Yield	Total Nitrogen	N-Uptake
	kg ha^−1^	%	kg N ha^−1^			kg N ha^−1^
NTC-PAC	3030 c	6.932 b	93.71 c	3240 d	7.03 b	101.57 d
CTC-PAC	2510 b	7.619 c	86.72 c	2650 c	7.59 c	91.79 c
NTC-PEC	2280 a	6.981 b	66.76 b	2410 b	6.95 b	71.31 b
CTC-PEC	2120 a	6.146 a	49.89 a	2230 a	6.04 a	52.81 a
LSD	59.7	0.1312	3.07	4.43	0.077	2.242

Different letters at each column denote statistically significant difference of means according to the LSD test for 95% significance level (*p* < 0.05).NTC-PAC: no tillage eplanting parallel to the contour direction. CTC-PAC: conventional tillage planting parallel to the contour direction. NTC-PEC: no tillage planting perpendicular to the contour direction. CTC-PEC: conventional tillage planting perpendicular the contour direction.

## Data Availability

Not applicable.

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
