# Peer review of "Effect of Different Tillage Practices on Sunflower (Helianthus annuus) Cultivation in a Crop Rotation System with Intercropping Triticosecale-Pisum sativum"

_plants, 2022, doi:10.3390/plants11243500_

Round 1
Reviewer 1 Report
Dear Editor-in-Chief,
Thank you for the opportunity to review this manuscript.
My observations follow below and should be carried out carefully.
In the title, the words Triticosecale and Pisum sativum must be written in italics
Line 21-22: The writing should be:
leaf area index (LAI), specific leaf area (SLA)
Line 32: The word Helianthus annuus must be written in italics.
Line 49-50: The word Helianthus annuus must be written in italics. Please make this adjustment and all parts of the text!
Line 58: The word Pisum sativum must be written in italics. Please, any word in Latin must be written in italics, be careful. This is a taxonomic rule.
Line 77: The word sativum must be written in italics.
Line 86: Please remove the dot between “Figure .1”
Line 88: 187 mm. The writing is separate!
Line 95: Standardize the script, Figure or Fig.
Line 106: In Figure 3 the decimal separator is wrong, it should be " . "
Line 129: In Figure 4 the decimal separator is wrong, it should be " . "
Line 164: it's Tables 3 and 4 not tables. The T is capitalized.
Line 165-166: What are the units of measure for values?
Line 221: The way of citing (Mujeeb-ul-Haq et al., 2020) is wrong, it must be numerical.
Line 227: the writing should be: in sunflower cultivation
Line 228: Please enter the units of measure in the LAI results
Line 249: the writing is: an altitude of 80 m above
Line 289: The writing should be: 60 kg P2O5 ha-1
Line 315: Please separate kg ha
Line 319: Were your plants dried at 700ºC?
Author Response
Dear Sir/Madam,
Thank you very much for reviewing my manuscript.
I send you my responses to your suggestions
Point 1: In the title, the words Triticosecale and Pisum sativum must be written in italics
Response 1: Corrected.
Point 2: Line 21-22: The writing should be: leaf area index (LAI), specific leaf area (SLA)
Response 2: Corrected.
Point 3: Line 32: The word Helianthus annuus must be written in italics.
Response 3: Corrected.
Point 4: Line 49-50: The word Helianthus annuus must be written in italics. Please make this adjustment and all parts of the text!
Response 4: Corrected.
Point 5: Line 58: The word Pisum sativum must be written in italics. Please, any word in Latin must be written in italics, be careful. This is a taxonomic rule.
Response 5: Corrected.
Point 6: Line 77: The word sativum must be written in italics.
Response 6: Corrected.
Point 7: Line 86: Please remove the dot between “Figure .1”
Response 7: Corrected.
Point 8: Line 88: 187 mm. The writing is separate!
Response 8: Corrected.
Point 9: Line 95: Standardize the script, Figure or Fig.
Response 9: Corrected.
Point 10: Line 106: In Figure 3 the decimal separator is wrong, it should be " . "
Response 10: Corrected.
Point 11: Line 129: In Figure 4 the decimal separator is wrong, it should be " . "
Response 11: Corrected.
Point 12: Line 164: it's Tables 3 and 4 not tables. The T is capitalized.
Response 12: Corrected.
Point 13: Line 165-166: What are the units of measure for values?
Response 13: Corrected.
Point 14: Line 221: The way of citing (Mujeeb-ul-Haq et al., 2020) is wrong, it must be numerical.
Response 14: Corrected.
Point 15: Line 227: the writing should be: in sunflower cultivation
Response 15: Corrected.
Point 16: Line 228: Please enter the units of measure in the LAI results
Response 16: Corrected.
Point 17: Line 249: the writing is: an altitude of 80 m above
Response 17: Corrected.
Point 18: Line 289: The writing should be: 60 kg P2O5 ha-1
Response 18: Corrected.
Point 19: Line 315: Please separate kg ha
Response 19: Corrected.
Point 20: Line 319: Were your plants dried at 700ºC?
Response 20: Corrected.
Reviewer 2 Report
This manuscript explored the physiological effects of different tillage practices in Sunflower (Helianthus annuus) cultivation in a crop rotation system with intercropping Triticosecale -Pisum sativum. Sunflower is a appropriate crop in a rotation system with intercropping Triticosecale x Pisum sativum for the rainfed sloping conditions. The experimental design is fine; however, the following points should be considered to improve this work:
Abstract
1. L17, “a two year” should be changed to “a two-year”
2. L18, “no tillage - planting” should be changed to “no tillage-planting”
3. L19, “tillage - planting” should be changed to “tillage-planting”
Introduction
4. L47, “helps” should read “help”
5. L62, “pea – maize and pea – sunflower” should be changed to “pea–maize and pea–sunflower”
Results
6. L98, “were observed” should be changed to “was observed”
7. Where is Figure 2?
8. L110, “leaf area index” should be changed to “Leaf area index”
9. L112, “15/9/2015” and “4/9/2016” should be changed to “and 15/9/2015” and “and 4/9/2016”
Discussion
10. L227, “applied” should be changed to “were applied”
11. L232, “the nitrogen values was” should be changed to “the nitrogen values were”
12. L236, “the 2nd year” should be changed to “the 2nd year”.
Materials and Methods
13. L255, “Soil sample” should be changed to “The soil sample”
14. L271, “A two – year field” should be changed to “A two–year field”
15. L278, “consisted of eight rows” should be changed to “consisting of eight rows”
16. L279, “sunflower” should be changed to “sunflowers”
17. L281, “at a rate 30% - 70%” should be changed to “at a rate of 30%-70%”
18. L310, “Specific leaf area Leaf area index” should be changed to “Specific leaf area and Leaf area index”
Conclusions
19. L345, “intecropping” should be changed to “intercropping”
Author Response
Dear Sir/Madam,
Thank you very much for reviewing my manuscript.
I send you my responses to your suggestions
Abstract
Point 1: L17, “a two year” should be changed to “a two-year”
Response 1: Corrected.
Point 2: L18, “no tillage - planting” should be changed to “no tillage-planting”
Response 2: Corrected.
Point 3: L19, “tillage - planting” should be changed to “tillage-planting”
Response 3: Corrected.
Introduction
Point 4: L47, “helps” should read “help”
Response 4: Corrected.
Point 5: L62, “pea – maize and pea – sunflower” should be changed to “pea–maize and pea–sunflower”
Response 5: Corrected.
Results
Point 6: L98, “were observed” should be changed to “was observed”
Response 6: Corrected.
Point 7: Where is Figure 2?
Response 7: Corrected.
Point 8: L110, “leaf area index” should be changed to “Leaf area index”
Response 8: Corrected.
Point 9: L112, “15/9/2015” and “4/9/2016” should be changed to “and 15/9/2015” and “and 4/9/2016”
Response 9: Corrected.
Discussion
Point 10: L227, “applied” should be changed to “were applied”
Response 10: Corrected.
Point 11: L232, “the nitrogen values was” should be changed to “the nitrogen values were”
Response 11: Corrected.
Point 12: L236, “the 2nd year” should be changed to “the 2nd year”.
Response 12: Corrected.
Materials and Methods
Point 13: L255, “Soil sample” should be changed to “The soil sample”
Response 13: Corrected.
Point 14: L271, “A two – year field” should be changed to “A two–year field”
Response 14: Corrected.
Point 15: L278, “consisted of eight rows” should be changed to “consisting of eight rows”
Response 15: Corrected.
Point 16: L279, “sunflower” should be changed to “sunflowers”
Response 16: Corrected.
Point 17: L281, “at a rate 30% - 70%” should be changed to “at a rate of 30%-70%”
Response 17: Corrected.
Point 18: L310, “Specific leaf area Leaf area index” should be changed to “Specific leaf area and Leaf area index”
Response 18: Corrected.
Conclusions
Point 19: L345, “intecropping” should be changed to “intercropping”
Response 19: Corrected.
Reviewer 3 Report
The Authors, in a 2-year experiment, evaluated the sunflower's reaction to two tillage systems (conventional and no-tillage) made parallei and perpendicular to the contour.
The authors evaluated selected plant parameters such as plant height, leaf area index (LAI), nitrogen and protein content in plants and nitrogen uptake by sunflower and Triticosecale/Pisum sativum. On the basis of these parameters, it is difficult to assess the purpose of the study, because the most important parameters, such as the yield of sunflower seeds and the oil content in the seeds, are missing.
Detailed notes:
- table 1 and 3 - results should be given with an accuracy of 0.01; the LSD value then enter up to 0.001 at most
- table 2 and 4 - results should be given with an accuracy of 0.01; Letter designations as "a" in Table 2 indicate no differences, while the difference between, for example, NTC -PAC is 0.135 and is more than 5 times higher than the value of LSD (0.0245); similar note apply to table 4
- why, however, the authors present the protein content in tables 3 and 4 since it is only a mathematical conversion of the nitrogen content from tables 1 and 2 (N content x 6.25 = protein content), for example: 4.388% N (table 1) x 6.25 = 27.43% protein (Table 3).
- how the nitrogen uptake was calculated in tables 5 and 6; lack of information on which organs of the sunflower it concerns (whole plants or stems after threshing the seeds), how was the plant biomass necessary to calculate nitrogen uptake. The designations of homogeneous groups in tables 5 and 6 also raise doubts - if the LSD is 6.4528, then in table 5 there should be the letters a, b, c and d and not a, b, bc and c; similar note to table 6.
- in general, the calculated LSD values are very low for the use of only 3 repetitions of the experiment
- table 9 - how nitrogen uptake was calculated - no data on plant biomass and nitrogen content, no statistical evaluation was given
- in materials and methods there is information: „total protein content was measured in the dry grains”- there is no such data in the manuscript
- in the context of inaccurately described research results, the presented discussion of the results is similarly very incomprehensible,
The work, in order to be processed further, requires a significant re-editing, a precise description of the parameters, and above all, supplementation with seed yield and oil content.
Author Response
Dear Sir/Madam,
Thank you very much for reviewing my manuscript.
I send you my responses to your suggestions
Point 1: table 1 and 3 - results should be given with an accuracy of 0.01; the LSD value then enter up to 0.001 at most
Response 1: the used statistical package Statgraphics plus 8.1 gives an accuracy of 0.05.
Point 2: table 2 and 4 - results should be given with an accuracy of 0.01; Letter designations as "a" in Table 2 indicate no differences, while the difference between, for example, NTC -PAC is 0.135 and is more than 5 times higher than the value of LSD (0.0245); similar note apply to table 4
Response 2: the used statistical package Statgraphics plus 8.1 gives an accuracy of 0.05. There was an error in the statistically results inadvertently. The error in manuscript was corrected.
Point 3: why, however, the authors present the protein content in tables 3 and 4 since it is only a mathematical conversion of the nitrogen content from tables 1 and 2 (N content x 6.25 = protein content), for example: 4.388% N (table 1) x 6.25 = 27.43% protein (Table 3).
Response 3: In our study we want to present both the total nitrogen and protein content results using the equation protein content = N content x 6.25.
Point 4: how the nitrogen uptake was calculated in tables 5 and 6; lack of information on which organs of the sunflower it concerns (whole plants or stems after threshing the seeds), how was the plant biomass necessary to calculate nitrogen uptake. The designations of homogeneous groups in tables 5 and 6 also raise doubts - if the LSD is 6.4528, then in table 5 there should be the letters a, b, c and d and not a, b, bc and c; similar note to table 6.
Response 4: the nitrogen uptake was calculated using the equation N-uptake = yield X plant’ total nitrogen (it is referred in Subsubsection 4.4.4.). The nitrogen uptake values concern the whole plant biomass. The statistic for the designation of homogeneous groups was conducted again and the results are as they referred in Tables 5 and 6.
Point 5: in general, the calculated LSD values are very low for the use of only 3 repetitions of the experiment
Response 5: the LSD values evaluated according to the statistical package
Point 6: table 9 - how nitrogen uptake was calculated - no data on plant biomass and nitrogen content, no statistical evaluation was given
Response 6: plant biomass, nitrogen content and statistical evaluation were added.
Point 7: in materials and methods there is information: „total protein content was measured in the dry grains”- there is no such data in the manuscript
Response 7: information changed
Point 8: in the context of inaccurately described research results, the presented discussion of the results is similarly very incomprehensible.
Response 8: changed have been done in discussion section
Point 9: The work, in order to be processed further, requires a significant re-editing, a precise description of the parameters, and above all, supplementation with seed yield and oil content.
Response 9: We did not calculate the oil content and the yield is published in another manuscript.
Round 2
Reviewer 1 Report
Dear Editor-in-Chief,
Thank you for making available a corrected version of this manuscript.
I am satisfied with the adjustments made.
All the best.
Author Response
Dear Reviewer,
Thank you very much for reviewing my manuscript.
Best Regards
Katerina Molla
Reviewer 2 Report
This version can be accepted.
Author Response

(The authors gave the same response as above.)

Reviewer 3 Report
The comments contained in the review have been taken into account by the Authors.
The authors are in favor of leaving tables 3 and 4, however they duplicate the data from tables 1 and 2 after multiplying by the value of 6.25 - the final decision should be made by the editor.
Regardless of the decision, the data in tables 1 and 3 in the "CV%" column require correction in this situation - they should be the same, but unfortunately they are different (table 1: 8.15; 3.22; 1.76; 6.15; table 2: 8.15; 1.76; 6.15; 3.22).
Author Response
Dear Reviewer,
Thank you very much for your valuable suggestions.
Point 1: The authors are in favor of leaving tables 3 and 4, however they duplicate the data from tables 1 and 2 after multiplying by the value of 6.25 - the final decision should be made by the editor.
Response 1: I will let the decision to be made by the editor
Point2: Regardless of the decision, the data in tables 1 and 3 in the "CV%" column require correction in this situation - they should be the same, but unfortunately they are different (table 1: 8.15; 3.22; 1.76; 6.15; table 2: 8.15; 1.76; 6.15; 3.22).
Response 2: corrected
Best Regards
Katerina Molla